# Preliminary Investigation of Sequential Application of Different Calcium Oxalate Solutions for Carbonate Rock Conservation

**Jianrui Zha [1], Yaoqi Gu [1], Shuya Wei [1,*], Huarui Han [2], Ankun Wang [3] and Qinglin Ma [4,*]**

[1]  Institute of Cultural Heritage and History of Science and Technology, University of Science and Technology Beijing, Beijing 100083, China
[2]  Institute of Archaeology, Chinese Academy of Social Sciences, Beijing 100101, China
[3]  Henan Provincial Institute for the Conservation of Architectural Heritage, Zhengzhou 450002, China
[4]  International Joint Research Laboratory of Environmental and Social Archaeology, Shandong University, Qingdao 266237, China
*   Correspondence: sywei66@hotmail.com (S.W.); qinglinma226@126.com (Q.M.); Tel.: +86-18701318128 (S.W.)

**Abstract:** Being inspired by nature, a series of experiments was carried out to deposit a calcium oxalate layer on the surface of the stone by the reaction between carbonate rock and oxalate salt. To increase the anti-dissolution properties of the calcium oxalate layer, the use of mixed oxalate solution has been proposed in the literature by two main routes: (1) adding acid agent to ammonium oxalate, which has the advantage of changing the particle structure and reducing layer porosity, and (2) using neutral methyl oxalate solution, which has the advantage of surface coverage due to slowly hydrolysis. In this study, we investigated the sequential application of ammonium oxalate, methyl oxalate, neutral mixed, and calcium acetate acid mixed solution. With this method, calcium carbonate and calcium oxalate solution can react inside the stone to reinforce it. The protective film's coverage area can then be increased using dimethyl oxalate neutral mixed solution, and the crystal morphology can be modified with calcium oxalate acid mixed solution. The anti-dissolution properties of the coating were investigated using both a custom-designed apparatus and a selective outdoor environment. The coating displayed good acid resistance properties at pH 2–4. After one year of exposure, the coating is firmly bonded with the stone.

**Keywords:** carbonate rock; sequential application; whewellite

## 1. Introduction

In China, a large part of stone cultural heritage has suffered from chemical dissolution, and the process is accelerated in polluted areas by the presence of nitrogen and sulfur oxides, which have a synergistically accelerating effect [1]. In order to reduce the effect of chemical weathering, coating interventions can be used, especially with outdoor stone artifacts with a high open porosity [2]. Various organic and inorganic chemicals have been proposed for the conservation of stone cultural heritage surfaces in order to improve the substrate water-repellent and anti-dissolution properties [3,4]. However, organic polymer materials were demonstrated to be less compatible with the inorganic stone cultural heritages, particularly carbonate rock [5]. In certain cases, inorganic protection coatings are more compatible with carbonate rock. Some hydroxides, such as calcium hydroxide and magnesium hydroxide, have already been utilized as consolidant because they can react with atmospheric carbon dioxide and form a compatible inorganic coating on the stone surface [6–10]. Additionally, inorganic coatings such as calcium oxalate and calcium phosphate have been confirmed to effectively inhibit the dissolution related phenomenon of stone monuments and temples in Europe [11,12]. Because it can be removed, calcium oxalate is receiving much more attention in the field of heritage protection than calcium phosphate [13]. The idea of

producing artificial calcium oxalate as a protective layer to provide carbonate rock with an acid-resistant surface originated in the nineteenth century and numerous methods have been developed [14]. It is notable that the calcium oxalate layer can be obtained by reacting different kinds of solution with calcium ions originating from the substrate [15]. The treatment consists of applying a poultice of ammonium oxalate, dimethyl oxalate, and oxalic acid to carbonate rock surface, with the purpose of forming a calcium oxalate monohydrate layer [16] (Equations (1)–(3)).

$$Ca^{2+} + C_2O_4{}^{2-} + H_2O \rightarrow CaC_2O_4 \cdot H_2O, \tag{1}$$

$$CH_3OOCCOOCH_3 + H_2O \rightarrow H_2C_2O_4 + CH_3CH_2OH, \tag{2}$$

$$H_2C_2O_4 + CaCO_3 \rightarrow CaC_2O_4 \cdot H_2O + CO_2, \tag{3}$$

Notably, unreacted solutions, other calcium oxalate phases (calcium oxalate dihydrate and metastable crystalline CaO$x$ phases), and reaction by-products (magnesium oxalate phase), all of which have the potential to influence the coating properties, remain at the end of the treatment, as confirmed by multiple techniques in previous studies [17,18]. According to researchers, a major portion of this problems were related to heterogeneous substrates. For example, carbonate rock comprises of calcite and dolomite. When dolomite contributes $Ca^{2+}$ and $Mg^{2+}$ ions to the coating-formation process, calcium and magnesium oxalate rims are produced, respectively. The high solubility of magnesium oxalate in water, and the restricted spatial distribution of the magnesium oxalate phase influence the coating coverage [19]. Therefore, a variety of oxalate salt solution combinations were investigated in an effort to prevent the described drawback [20].

The advantages of the oxalate salt mixed solution were initially proposed in the modification of crystal structure and afterwards assessed in the extension of the coating coverage area [21]. Current research on marble and limestone samples has shown that acid solution treatments affect whewellite crystal preferred orientations. This leads to a more homogenous crystal structure, which reduces porosity [22]. Moreover, studies on marble found that adding neutral diethyl oxalate solution can dramatically increase surface coverage while lowering cracking and porosity. This is because the molecule slowly hydrolyzes to minimize ionic interaction throughout the process of dissolution and participation [23]. This, in turn, increases the resistance of the samples against acid dissolution. If the solutions described above could work together simultaneously, the protective effect of the calcium oxalate layer would be enhanced. However, in the absence of acid, the hydrolysis rate of methyl oxalate increases, and the resulting product oxalic acid loses the advantage effect [24]. This remains a big challenge.

The calcium phosphate layer deposition experiment may provide inspiration for conservators seeking solutions to their issues. In a previous study, it was discovered that application of a limewater poultice after treatment with diammonium hydrogen phosphate increased calcium phosphate formation and removed the unreacted agent [25]. However, when ammonium oxalate and diammonium hydrogen phosphate are applied sequentially, a patchy layer develops [26]. This is due to the fact that whewellite is formed when ammonium oxalate reacts with calcite, which reduces the stone's porosity. As a result, when the diammonium hydrogen phosphate solution is applied, there are no more calcium ions on the surface to form calcium phosphate, and deep penetration of the diammonium hydrogen phosphate solution into the substrate is inhibited. So, layer deposition can be improved by the sequential application of calcium gel or micro-grout solutions.

Therefore, in this research the sequential application of ammonium oxalate, methyl oxalate neutral mixed and calcium acetate acid mixed solution on dolomite marble was investigated. To evaluate the effect, firstly, the chemical composition and structure morphology of the deposition layer were characterized. The resistance to dissolution was assessed by being exposed to specific outdoor environments as well as a custom apparatus that could simulate rain, as in the references [26]. Through the above dissolution resistant experiment, it is possible to find the effects of the calcium oxalate layer, which provides

direction for the sequential application method. This research concentrates on the study of coating densification and crystal regulation, while other research gives more attention to the preparation method, application environment, and performance evaluation [6,27].

## 2. Materials and Methods

### 2.1. Coating Formulation

For the coating preparation, ammonium oxalate $((NH_4)_2C_2O_4)$, calcium acetate $(Ca(CH_3COO)_2 \cdot H_2O)$, oxalic acid $(H_2C_2O_4)$, nitric acid $(HNO_3)$, dimethyl oxalate $(C_4H_6O_4)$, sodium hydroxide $(NaOH)$, and ethanol $(C_2H_5OH)$ were used. All of the chemicals were acquired from Aladdin-reagent Co., Ltd. (Shanghai, China) and without further purification. Three distinct solutions were obtained as previous report: 0.1 mol/L ammonium oxalate solution $((NH_4)_2C_2O_4 + H_2O)$, 0.02 mol/L Calcium oxalate solution A $(Ca(CH_3COO)_2 \cdot H_2O + C_4H_6O_4 + NaOH + H_2O + C_2H_5OH)$, and 0.025 mol/L Calcium oxalate solution B $(Ca(CH_3COO)_2 \cdot H_2O + H_2C_2O_4 + HNO_3 + H_2O)$. To obtain the best results when applied to outdoor stone, the entire drying period of the solution was set at more than 12 h.

As depicted in Figure 1, obtained solutions were sequentially applied to the selected substrate: one application of ammonium oxalate solution, four applications of Calcium oxalate solution A, and one application of Calcium oxalate solution B. Coatings were deposition on a modest substrate (25 mm × 25 mm × 10 mm) for laboratory analysis, and on a natural stone for outdoor exposure. The coating was cured by drying for 24 h at room temperature until a consistent mass was observed.

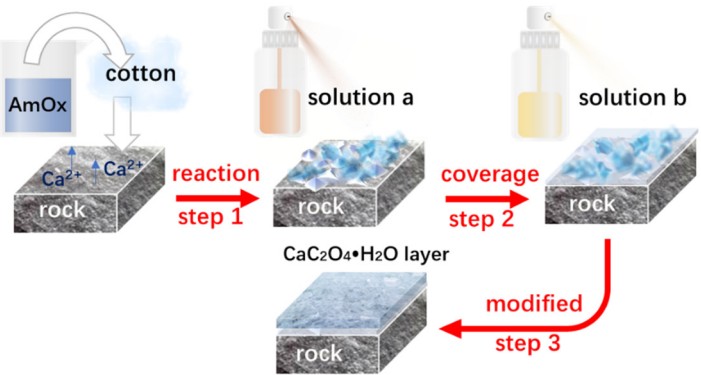

**Figure 1.** Scheme illustrating the treatment process of each sample.

### 2.2. Stone Substrate

Two kinds of calcareous stone substrates were selected to evaluate the composition, morphology, and performance of the coating. Modest marble substrate (Angera stone) with dimensions of 25 mm × 25 mm × 10 mm was obtained from the local market and polished with SiC paper (180#). Nature stone (green-white stone) was collected from the Fangshan district, where a large quantity of stone relics had been unearthed. Both stones were ultrasonic cleaned with deionized water and baked for 24 h at 110 °C in order to remove the dust on the surface.

### 2.3. Characterization Techniques

X-ray diffraction (XRD, Bruker D8 Advance, Karlsruhe, Germany) analysis was used to analyze the coated stone and acid rain attacked samples. The X-rays were generated with Cu Kα radiation at 45 kV and 200 mA in a scanning range of 10–70(2θ) with a scanning speed of 1°/min and step size of 0.02°. Before measurements, the samples were cut to meet the requirements of the analysis.

Raman spectrum were performed by using a Horiba Scientific LabRAM HR confocal Raman spectrometer (Horiba Scientific, Lille, France). The coated stone, the acid rain attacked samples and the outdoor exposure samples were excited with 785 nm laser, using the 50× objective lens with 1 μm spatial resolution. In order to improve the signal to noise

ratio, the acquisition time was fixed at 60 s with 1 accumulation. The analyses performed on surface sections placed on the motorized X-Y microscope stage. Raman spectra were recorded automatically along an area. A great advantage of this method is its high spatial resolution (4–5 μm). In addition, a visible-light microscope allows characterized of a selected area. So, the distribution of the phases was obtained by area maps.

Fourier transform infrared spectra were collected on FTIR spectrophotometer, Thermo Fisher Nicolet (Waltham, MA, USA) is 5, in the range of 500–4000 cm$^{-1}$, using resolution 4 and 16 scan. The coated marble substrate was done using attenuated total reflection (ATR) accessory.

Scanning electron microscope (SEM) images were obtained using Tescan Vega3 (Brno, Czech Republic) from surface of coated stone, the acid rain attacked samples, the outdoor exposure samples, and cross-sections of coated stone as well as the outdoor exposure samples. All of the samples were sputter-coated with gold, and observed at an accelerating voltage of 20 kV.

The surface roughness of coated stone was imaged in the tapping mode in air using Bruker Nano Dimension Edge AFM (Bruker, Billerica, MA, USA). Samples were calculated at controlled room temperature (22 °C).

The acid resistance of the specimens was evaluated by simulating the rain dissolution process learn from Graziani reference [26]. Acid solution composed of $HNO_3$ and $H_2SO_4$ (1:1, pH = 1, 2, 3, 4) was dripped onto the coated or uncoated marble samples. Furthermore, the coated green-white stone began to be exposed outside in February of 2021. After one year of exposure, the environmental degradation of the sample was evaluated. Table 1 summarizes the characteristics of the preceding samples.

**Table 1.** Characteristics of the samples.

| Sample | Type | Test Description | Analysis Description |
|--------|------|-----------------|---------------------|
| 1 | coated stone | according to analysis requirement | XRD, Raman, FTIR, AFM, SEM (surface and cross section) |
| 2 | stone-1 | simulate acid rain dissolution experiment at pH 1 | XRD, Raman, SEM (surface) |
| 3 | stone-2 | simulate acid rain dissolution experiment at pH 2 | XRD, Raman, SEM (surface) |
| 4 | stone-3 | simulate acid rain dissolution experiment at pH 3 | XRD, Raman, SEM (surface) |
| 5 | stone-4 | simulate acid rain dissolution experiment at pH 4 | XRD, Raman, SEM (surface) |
| 6 | coated stone-1 | simulate acid rain dissolution experiment at pH 1 | XRD, Raman, SEM (surface) |
| 7 | coated stone-2 | simulate acid rain dissolution experiment at pH 2 | XRD, Raman, SEM (surface) |
| 8 | coated stone-3 | simulate acid rain dissolution experiment at pH 3 | XRD, Raman, SEM (surface) |
| 9 | coated stone-4 | simulate acid rain dissolution experiment at pH 4 | XRD, Raman, SEM (surface) |
| 10 | coated green-white stone | exposure to outside environment (Beijing huairou district) | Digital image, Raman mapping, SEM (surface section) |

## 3. Results and Discussion

### 3.1. Characterization of the Film

As previously stated, it is well known that the application of ammonium oxalate solution can form a $CaC_2O_4 \cdot H_2O$ coating, thereby inhibiting the degradation of the stones [28]. In order to determine the influence of combined treatment with ammonium oxalate solution, calcium oxalate solution A, and calcium oxalate solution B, the coated marble stones' composition was thoroughly investigated.

First of all, XRD studies of the crystalline phases (Figure 2a) reveals the corresponding Bragg peaks for $CaMg(CO_3)_2$ and $CaC_2O_4 \cdot H_2O$ in the coated stone. It is not surprising that weddellite and glushinskite are missing under the combined treatment. In fact, a number of studies discovered that only whewellite precipitated at pH < 5, whereas glushinskite was absent from the treated dolomite marble due to low amount of this phases in the reaction rim [29]. Moreover, the Raman spectra (Figure 2b) of the coated stone reveals that the peaks of whewellite at 140, 192, 246, 501, 597, 725, 895, 1462, 1489 cm$^{-1}$ were almost not shifted to the other wavenumbers [30]. This result may be attributed to the absence

of amorphous magnesium oxalate, which is usually detected in dolomite treated with ammonium oxalate solution. After the combined use of different calcium oxalate solution and subsequent washing, the possibility amorphous magnesium oxalate was washed out. Moreover, FTIR reveals a weak Fe–O bond at 585 cm$^{-1}$, with the exception of the whewellite group (Table 2). The associated mineral still present thanks to the lower impact of ammonium oxalate solution and neutral pH value of the calcium oxalate solution A [31].

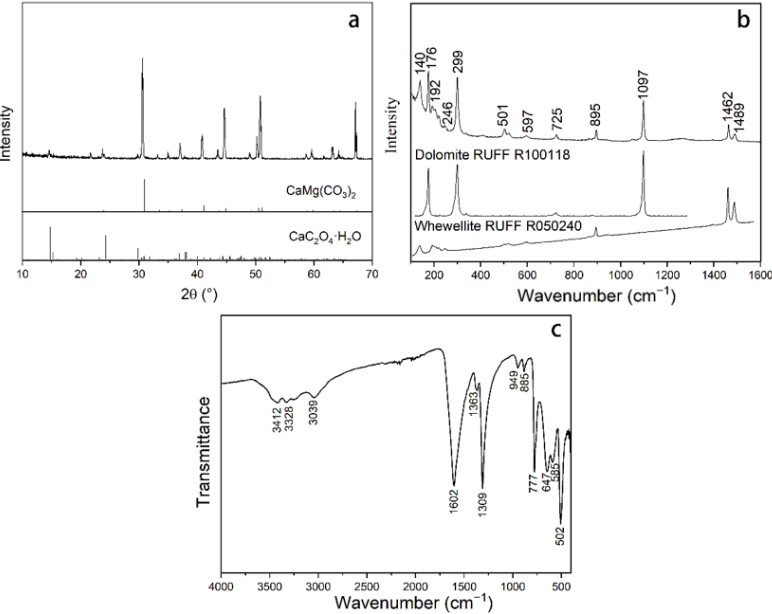

**Figure 2.** Composition of treated samples. (**a**) XRD, (**b**) Raman, (**c**) FTIR.

**Table 2.** 2θ degree of test samples.

| Samples | 2θ (°) | hkl |
| --- | --- | --- |
| Dolomite (36–0426) | 30.938 | 104 |
| Coated Stone-1 | 30.937 | 104 |
| Coated Stone-2 | 30.960 | 104 |
| Coated Stone-3 | 30.982 | 104 |
| Coated Stone-4 | 30.978 | 104 |
| Stone-1 | 30.880 | 104 |
| Stone-2 | 30.940 | 104 |
| Stone-3 | 31.000 | 104 |
| Stone-4 | 30.940 | 104 |

In order to investigate the morphology of the newly formed crystals, the same portions of the treated samples analyzed by Raman spectra were observed by scanning electron microscope. The sample surface is completely coated with calcium oxalate as observed in Figure 3a. The crystals of whewellite are grouped in agglomerates with an appreciable tabular structure. The sizes of many crystals are around 1 µm. Interestingly, a homogenous calcium oxalate layer without distinct layer structure is observed on the dolomite marble surface after combined treatment (Figure 3b); also, etch pits are not visible on the substrate surface. Furthermore, the inner layer between the unreacted substrate and the coating has a comparable texture to the surface layer. It is unclear if this is the penetration result of the solution A or solution B, or a consequence of the combined deposition process. These results are consistent with AFM morphological analysis (Figure 3c). The surfaces of stone are covered with tabular clusters of uniform size and low standard deviations for both particle species. Moreover, pseudocolor pictures indicate the depth field of surface features, in which bright color represents the elevated features, and darker colors represent deeper ones. Variation in the topographical image of the surface indicates the roughness of the

whole sample. However, considering the roughness structure of dolomite basement, these roughness values are relatively lower than those of traditional treatment, as shown in Figure S1 (see Supplementary Materials).

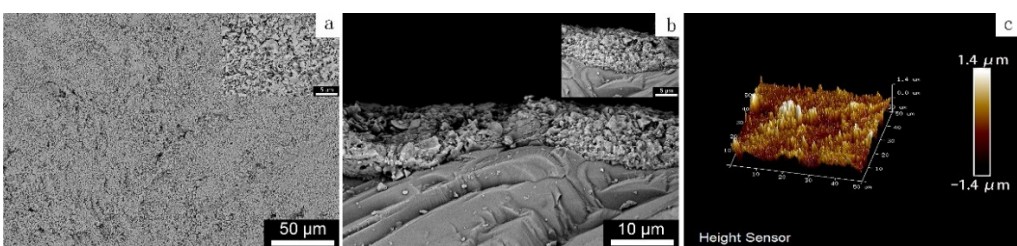

**Figure 3.** Morphology of treated samples. (**a**) SEM-surface section, (**b**) SEM-cross section, (**c**) AFM-surface section.

### 3.2. Protective Properties of the Film

Stone and coated stone were subjected to a dripping test at pH 1, 2, 3, 4 to evaluate their resistance to simulate acid rain dissolution. Lower pH values were selected in order to reproduce severe conditions and observe the specimens' deterioration. In fact, the dissolution process of calcium oxalate is related to ionic strength, protonation reactions, and the production of soluble complex [32]. The laboratory experiments focus on the dissolution mechanism, which is the most important part of the dissolution process. Considering the changeable outdoor environment, a field aging test should also be conducted on the coated samples to evaluate the protective properties.

Regarding the dissolution appearance of stone, the XRD pattern (Figure 4a) exhibits the dolomite diffraction peaks around 30.9°, 35.3°, 37.4°, 41.1°, 44.9°, 51.1°, 58.9°, and 67.4°, corresponding to (104), (015), (110), (113), (202), (116), (211), and (300) preferred orientations, respectively. These values are consistent with the international Center of Diffraction Data (ICDD) card number (36-0426). The strongest peak occurs at 2θ~30.9°, which is referred to (104) plane. The position of the peaks shifted after simulating acid rain dissolution, and the coated stone was altered at pH 1–2, leading to the conclusion that the calcium oxalate layer was destroyed during the dissolution process. In addition, it can be noticed that 2θ for stone changes irregularly, and maybe other impure constituents affects the result. The absence of other aging products suggests that the $CaSO_4$ or $Ca(NO)_3$ crystals are easily washed out with runoff solution. In Figure 4b, we compare the Raman spectra of stone and coated stone treated with acid rain. The main peak of dolomite is not shifted, and two samples (coated stone-3 and coated stone-4) exhibit prominent whewellite peaks when the pH value of simulated acid solution is 3 and 4. An obvious rise in dolomite peak width (FWHM) at higher wave numbers is detected at Coated stone-2, Coated stone-1, and stone-1. The feature is the result of the dissolution of other crystals. Therefore, the stone and the coated stone undergo two different dissolution processes.

In order to better characterize the sample dissolution process, the surface morphologies of all samples after simulated acid rain dissolution are shown in Figure 5. On the stone surfaces, cracks of varying degrees emerge, signifying the different levels of degradation correlating to the pH value of acid solution. In comparison, coated stone exhibits vast changes after simulated acid rain dissolution. The surface denudation of the sample treated with pH 1 solution confirms that the sample was severely damaged. On the surface of the other treated sample, only little holes are presented, implying stronger dissolution resistance. It is possible that the uniform calcium oxalate layer inhibited the dissolution process on the stone's surface. The detachment of tightly bonded coating promotes the destruction of substrate. In order to better reflect the performance of this treatment, we compared the results with our previous studies using part components of the chemicals, including using ammonium oxalate individually or together with solution A or solution B. All the corresponding SEM images are shown in Figures S2 and S3 (see Supplementary

Materials). This comparison can prove that the coating formed through this treatment is more uniform.

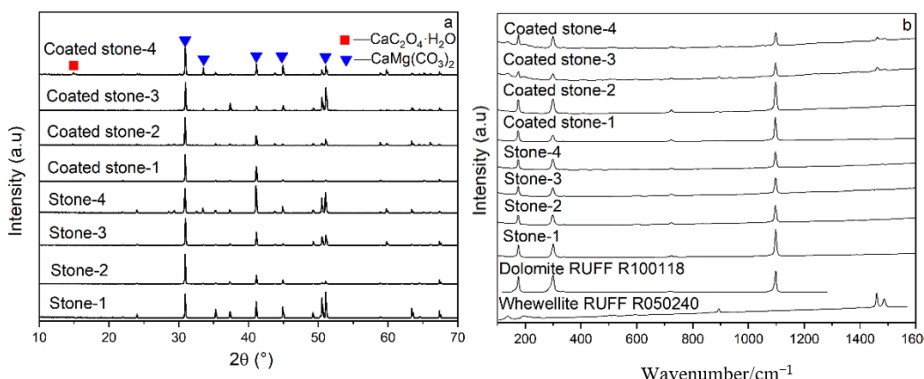

**Figure 4.** Composition of test samples after treatment with simulated acid rain at different pH values. (**a**) XRD, (**b**) Raman.

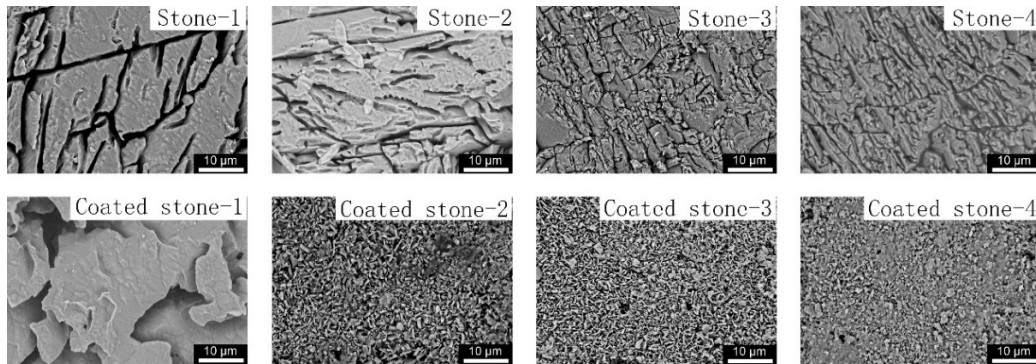

**Figure 5.** SEM microscope of test samples after treatment with simulated acid rain at different pH values.

As mentioned in the introduction, the deterioration process of stone is influenced by the chemical, physical, and biological factors. Field exposure tests give the possibility to evaluate the environmental impact on the applied conservation actions. So, treated samples were subjected to exposure tests in typical environmental at a house estate in Beijing's Huairou District. There is no obvious change to the surface of green-white stone samples coated with calcium oxalate coating materials (Figure 6). The slight yellowing of the treated samples after 3 months is associated with dust deposition and the pollutant will be washed away by rain.

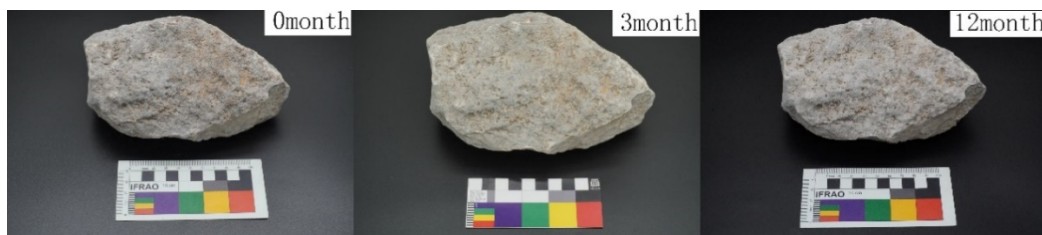

**Figure 6.** Field exposure results after one year.

Micro scale surface section of the field exposure sample was investigated with Raman spectral area map and SEM (Figure 7). Obvious evidence of the existence of the calcium oxalate was found; more specifically, the lightness of the red area correlates with the amount of calcium oxalate. After the field exposure, calcium oxalate is the dominant phase

of sample surface, small quant of dark area often related to the surface roughness of sample. As shown in Figure 7c, calcium oxalate crystals of different sizes accumulate on the rough sample surface, and porosities are rarely present. The results show the effectiveness and durability of this treatment method for practical outdoor stone conservation.

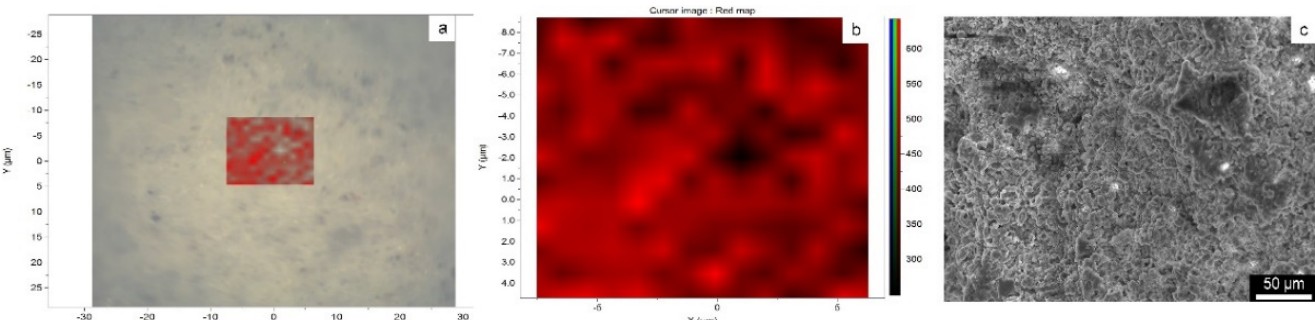

**Figure 7.** Field exposure results after one year. (**a**) Micrograph, (**b**) Raman mapping, (**c**) SEM.

### 4. Conclusions

In this study, the sequential application of ammonium oxalate, methyl oxalate neutral mixed, and calcium acetate acid mixed solutions was investigated as a possible route to provide carbonate rock with acid resistance properties and durable conservation at the same time.

The resulting coating exhibits a homogenous and refined microstructure, presumably thanks to the slowly hydrolysis of methyl oxalate neutral mixed and the effect of calcium acetate acid mixed crystal on crystal structure modification. After prolonged exposure to simulated acid rain (pH 2–4) and a selective outdoor environment, the acid resistance properties of the coated stone exhibit greater durability. This is possible due to the bonding force between the previous deposition particle and the sequential deposition layer. In this way, sequential deposition particles are firmly incorporated into the previous layer, thereby forming a durable conservation layer.

Considering the durability of the coating investigated in this study, the advantage of the sequential application is evident. More experiments to assess the mechanism of the coating formation process will be the next step in the research.

**Supplementary Materials:** The following supporting information can be downloaded at: https://www.mdpi.com/article/10.3390/coatings12101412/s1, Figure S1: The AFM of dolomite marble after ammonium oxalate solution treatment; Figure S2: SEM images of (a) Marble, (b) Sample 'SF', (c) Sample 'PF' after immersion tests for 6 h; Figure S3: SEM images after UT (a,d), AO (b,e) and 0.02 mol/L CA (c,f) were treated with 750 mL acid solution($HNO_3$:$H_2SO_4$ = 1:1), with an initial pH of 4.0.

**Author Contributions:** Conceptualization, J.Z.; funding acquisition, S.W. and Q.M.; methodology, J.Z. and Y.G.; resources, S.W.; software, Y.G.; writing—original draft, J.Z.; writing—review and editing, H.H. and A.W. All authors have read and agreed to the published version of the manuscript.

**Funding:** This research was funded by Beijing Municipal Administration of Cultural Heritage (PXM2020-039208-000005).

**Institutional Review Board Statement:** Not applicable.

**Informed Consent Statement:** Not applicable.

**Data Availability Statement:** Not applicable.

**Acknowledgments:** The authors want to thanks Beijing Stone Carving Art Museum for the supply of material support in this work.

**Conflicts of Interest:** The authors declare no conflict of interest.

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
