# Peer review of "Preliminary Investigation of Sequential Application of Different Calcium Oxalate Solutions for Carbonate Rock Conservation"

_coatings, doi:10.3390/coatings12101412_

Round 1
Reviewer 1 Report
The proposed approach might be of interest to COATINGS, but some major revisions are necessary:
English form: please have it revised by a native speaker
Both abstract and intro are confusing: there are several chemical and physico-chemical processes mentioned, but altogether explanations are fuzzy and hard to follow.
Abstract and intro should be revised by:
- Giving more linear and clear explanations of the processes/reactions, using chemical equations/reaction schemes as much as possible to clarify the different interconnected processes
- Stress much better the novelty of the study as compared to state-of-the-art research on oxalates for the consolidation of carbonate materials
- In addition, regarding state-of-the-art methods, only oxalates and phosphates are mentioned, but there is no reference to the use of hydroxides (e.g. calcium and magnesium), which are widely accepted consolidants for these materials; the authors should at least include them in the range of possible inorganic consolidants around line 36-40, and several references can be found in the literature.
Methods and Discussion
The main issue here is that there is no clear comparison of the complete methodology with other applications where only some of the components of the total mixture were applied. This would be necessary to better assess and demonstrate the efficacy of the new methodology.
Author Response
Response to Reviewer 1 Comments
Point 1: English form: please have it revised by a native speaker.
Response 1: Thanks for providing us with this great opportunity to submit a revised version of our manuscript. We appreciate the detailed and constructive comments provided by you. We corrected the language problem as you suggested.
Point 2: Giving more linear and clear explanations of the processes/reactions, using chemical equations/reaction schemes as much as possible to clarify the different interconnected processes.
Response 2: In order to clarify the interconnected processes of applying ammonium oxalate, methyl oxalate neutral mixed and calcium acetate acid mixed solution to carbonate rock surface, we have added a description in P1 lines 21-24, and added corresponding reaction equations in P2 line 54.
Point 3: Stress much better the novelty of the study as compared to state-of-the-art research on oxalates for the consolidation of carbonate materials
Response 3: Most of the state-of-the-art researches are related to the preparation process, application environment and performance evaluation. Our research is more related to coating densification and crystal regulation, which is different from other studies. The discussion has been added on P3 lines 99-102.
Point 4: In addition, regarding state-of-the-art methods, only oxalates and phosphates are mentioned, but there is no reference to the use of hydroxides (e.g. calcium and magnesium), which are widely accepted consolidants for these materials; the authors should at least include them in the range of possible inorganic consolidants around line 36-40, and several references can be found in the literature.
Response 4: The method of using hydroxides as consolidats has been added on P1 lines 40-43, with the corresponding references.
Point 5: The main issue here is that there is no clear comparison of the complete methodology with other applications where only some of the components of the total mixture were applied. This would be necessary to better assess and demonstrate the efficacy of the new methodology.
Response 5: We've done some previous studies applying only some of the components of the total mixture,as just using ammonium oxalate、using ammonium oxalate with solution A,and using ammonium oxalate with solution B. The results have been published in other papers, so we have added a comparative discussion on P7, lines 247-252, and added relevant information in the Supplementary Materials.

Reviewer 2 Report
I found the manuscript "Preliminary Investigation of Sequential Application Different Calcium Oxalate Solution for Carbonate Rock Conservation" interesting and important. This development is significant for the preservation of cultural heritage monuments made of stone. This development is all the more interesting because the production of oxalic acid by organisms is often considered as a factor in the destruction of marble monuments. This work is aimed at preserving cultural heritage sites using oxalate coatings.
Experimental data showed good acid resistance of this coating, which is an important property of this coating.
The article is well written and illustrated, performed at a good methodological level.
In my opinion, it is very concise and contains little factual material, and the data obtained are practically not discussed in comparison with the literature. Perhaps this is due to the fact that there is really very little data on this problem.
Nevertheless, I would recommend the authors to expand the discussion of the results with the involvement of literary sources. At the moment, the list of references has only 26, which is very small.
Author Response
Response to Reviewer 2 Comments
Point 1: In my opinion, it is very concise and contains little factual material, and the data obtained are practically not discussed in comparison with the literature. Perhaps this is due to the fact that there is really very little data on this problem.
Response 1: Thank you very much for your time involved in reviewing the manuscript. We appreciate your clear and detailed feedback and hope that the explanation has fully addressed all of your concerns. We've done some previous studies applying only some of the components of the total mixture,as using ammonium oxalate individually、using ammonium oxalate with solution A together,and using ammonium oxalate with solution B together. The results have been published in other papers, so we added a comparative discussion on P7, lines 247-252, and added relevant information in the Supplementary Materials.
Point 2: Nevertheless, I would recommend the authors to expand the discussion of the results with the involvement of literary sources. At the moment, the list of references has only 26, which is very small.
Response 2: According to your suggestion, we have added more relevant references and compared them with the work in this article. The method of using hydroxides as consolidats has been added on P1 lines 40-43, with the corresponding references. And we have stressed much better the novelty of the study as compared to state-of-the-art research on oxalates for the consolidation of carbonate materials. Most of the state-of-the-art researches are related to the preparation process, application environment and performance evaluation. Our research is more related to coating densification and crystal regulation, which is different from other studies. The discussion has been added on P3 line 99-102.

Round 2
Reviewer 1 Report
the authors have addressed some of the reviewers' criticisms